# Association of Nutritional Factors with Hearing Loss

**DOI:** 10.3390/nu11020307

**Published:** 2019-02-01

**Authors:** Su Young Jung, Sang Hoon Kim, Seung Geun Yeo

**Affiliations:** 1Department of Otorhinolaryngology-Head and Neck Surgery, Myongji Hospital, Hanyang University, College of Medicine, Goyang 10475, Korea; monkiwh35@hanmail.net; 2Department of Otorhinolaryngology—Head & Neck Surgery, School of Medicine, Kyung Hee University, Seoul 02447, Korea; hoon0700@naver.com

**Keywords:** nutrition, diet, hearing, hearing loss

## Abstract

Hearing loss (HL) is a major public health problem. Nutritional factors can affect a variety of diseases, such as HL, in humans. Thus far, several studies have evaluated the association between nutrition and hearing. These studies found that the incidence of HL was increased with the lack of single micro-nutrients such as vitamins A, B, C, D and E, and zinc, magnesium, selenium, iron and iodine. Higher carbohydrate, fat, and cholesterol intake, or lower protein intake, by individuals corresponded to poorer hearing status. However, higher consumption of polyunsaturated fatty acids corresponded to better hearing status of studied subjects. In addition to malnutrition, obesity was reported as a risk factor for HL. In studies of the relationship between middle ear infection and nutrition in children, it was reported that lack of vitamins A, C and E, and zinc and iron, resulted in poorer healing status due to vulnerability to infection. These studies indicate that various nutritional factors can affect hearing. Therefore, considering that multifactorial nutritional causes are responsible, in part, for HL, provision of proper guidelines for maintaining a proper nutritional status is expected to prevent some of the causes and burden of HL.

## 1. Introduction

Hearing loss (HL), caused by partial or complete dysfunction of the auditory pathway from the external ear to the cerebral auditory cortex, is the most important and frequent symptom of ear disorders [1]. HL is a major public health problem and has been recently ranked as the fifth leading cause of years lived with disability, even higher than many other chronic diseases such as diabetes, dementia, and chronic obstructive pulmonary disease. Moreover, according to the World Health Organization (WHO), the incidence of auditory disorders is increasing at an alarming rate without much concern or perception by society and public health officials [2,3,4,5]. Because HL affects the quality of life and communication and relationships with others, it is related to the social costs as well as economic costs of medical treatment [6]. In particular, HL affects talking and has been associated with depression and anxiety [7]. Affected children are likely to experience delays in development of speech, language and cognition, and poor school performance [8], whereas adults face higher risks of unemployment or low earnings [9]. Recently, it has been reported that hearing impairment is associated with increased cognitive dysfunction and dementia in the elderly [10,11,12] and that social isolation accompanies the daily life of those with impaired hearing [13]. 

HL is most often a sensorineural origin due to irreversible loss of hair cells and/or spiral ganglion neurons. Causes of sensorineural HL (SNHL) are multifactorial factors, including genetic and environmental factors such as noise, toxicity and aging [14]. Sex has commonly been a statistically significant factor, with males having worse hearing thresholds than females [15]. Other statistically significant covariates include education/socioeconomic status [16,17], noise exposure [18], race/ethnicity [19], smoking and second-hand smoke [20,21], cardiovascular health [18,22], diabetes [23], genetics [24,25], and ototoxic drugs [23]. In addition to these factors associated with HL, severe prenatal iodine deficiency has been listed by the WHO as a nutritional cause of HL [26,27], leaving the broad roles of diet and nutrition within this complex set of etiologies yet to be defined. Nutritional status is the key to aging disability, but the interaction between nutrition and SNHL has only recently gained attention. Nutrition is the science of interpreting the interaction of nutrients and other substances in food with respect to the maintenance, growth, reproduction, health and disease of organisms. It includes food intake, absorption, assimilation, biosynthesis, catabolism and excretion [28]. Therefore, nutritional status or nutritional factors can affect various diseases occurring in humans. It is known that various diseases of the metabolic syndrome are affected by these nutritional factors, and HL may also be involved. To reduce the burden on HL, it is important to identify the protection factors and avoid risk factors. In this respect, nutritional factors could help the first step for the prevention and potential repair of hearing damage before it can reach an irreversible state. Therefore, it is important to closely review the nutritional factors related to the pathophysiologic mechanism of HL.

This review of studies of nutrition and hearing has been conducted to investigate the types of nutrients, other than the well-known toxic substances, that play an important role in HL and to determine the mechanisms that affect HL.

## 2. Evidence that Nutrition is a Factor Affecting HL

The results of these numerous previous studies revealed that the common causes of HL are age and noise exposure. The prevalence of HL is rapidly increasing due to population aging, increased loud environments, and increased use of listening devices [29]. Other major causes of this burden vary across the different stages of life and include congenital disorders, otitis media, ototoxic drugs, and vaccine-preventable infections such as measles, mumps, and rubella [29]. However, there is no definitive study on what roles nutrition plays in the pathophysiologic mechanism of HL. One possibility is that dietary quality influences hearing status by mediating vulnerability of the inner ear to age-related changes [30]. One of the factors associated with age-related change is free radical formation. Free radical formation in the inner ear is a key mechanism for HL [31,32], causing vasoconstriction leading to the death of the inner ear cells. Subsequent reperfusion of cochlear cells further contributes to free radical formation and further cell death, similar to what is observed in stroke. In addition, antioxidants such as vitamins, which inhibit the formation of free radicals, may play a specific role in preventing and treating HL [33]. Therefore, several studies have been reported on the relationship between HL and vitamins A [33,34,35,36,37,38,39], C [33,35,36,37,40], and E in humans [33,34,35,36,37,40]. Magnesium (Mg) has also been reported to reduce HL through synergistic effects with vitamins [33,36,41,42,43]. These findings suggest that free radical scavengers, such as vitamins A, C and E, act in synergy with Mg to reduce changes in hearing thresholds more reliably than treatment with any single agent. Therefore, higher intake of antioxidants and/or magnesium may be associated with a lower risk of HL. Selenium (Se) may also play a role in hearing [44,45]. Moreover, a lack of vitamin B has been reported to increase the risk of HL [39,46,47,48], whereas studies in animal models indicate that antioxidants reduce potential free radical damage associated with micronutrient deficiency. Exposure to intense noise over time activates morphological, physical and mechanical mechanisms that can cause cochlear damage and result in HL [31]. Metabolic and molecular mechanisms may also be responsible for this type of lesion [32]. Although the exact mechanisms required for the induction of temporary HL are unclear, recent studies indicate that the formation of reactive oxygen species (ROS) and oxidative stress are the main metabolic causes of temporary HL [43,48]. Permanent HL of cochlear origin is thought to be caused by the death of hair cells, both internal and external, primary afferent neurons, or both [41,42]. ROS are molecules characterized by the presence of unpaired electrons; they are always present in the body and participate in homeostasis and important signaling pathways. However, imbalances in the endogenous antioxidant system can increase ROS concentrations to toxic levels, causing cell death due to damage to cell membranes, cytoplasm and mitochondria. In particular, excess free radicals in the cells of the cochlear sensory epithelium, spiral ganglion neurons and cochlear blood vessels can play an important role in the onset of HL. Excess ROS, induced by intense noise exposure and toxic drugs, is a key factor in the pathogenesis of HL, along with other stresses and aging [33,35,36,37,44,45]. Several studies have shown that loud noise induces unnecessary ROS production and the overproduction of certain cells in the cochlea resulting in cell damage and noise-induced HL. Under these conditions, high blood concentrations of antioxidants could inhibit the production of ROS, suggesting that antioxidants reduce cell damage by suppressing ROS production in hair cells exposed to noise [31,32,33].

In contrast to vitamins and minerals, higher carbohydrate intake [49], fat intake [36,50,51,52,53], and cholesterol intake [54,55,56] have been reported to have a negative effect on hearing outcome. Damage to the blood supply to the cochlea can contribute to a decrease in auditory sensitivity, and thus vascular factors are known to play a key role in the development and progression of HL [57]. Excessive consumption of carbohydrates, fats and cholesterol can increase the risk of cardiovascular [58] and cerebrovascular disease [59], and similar mechanisms can damage cochlea blood flow [60]. In contrast, consumption of fish and higher intakes of long-chain omega-3 polyunsaturated fatty acids (PUFAs) have been reported to have a positive impact on auditory sensitivity by improving blood supply to the cochlea [61]. Taken together, these results suggest that micronutrients and macronutrients can interact in a variety of forms on the hearing outcome.

In another study, a Healthy Eating Index (HEI) was used as a measure of total dietary value to utilize dietary patterns as an alternative method to investigate the relationship between nutrition and hearing considering the limitations of single nutritional analysis [30,62]. Other studies have also used biomarkers (such as serum albumin levels) or anthropometric indices to assess general nutritional status [63]. All reported that the general nutritional status was a factor that could affect HL and that the risk of HL was increased with poorer general nutritional status.

There are also studies on the relationship between pediatric HL and nutrition [64,65,66,67,68]. According to these studies, certain nutritional deficiencies associated with iodine and thiamine can increase the risk of HL, even with only limited data and an unquantified threshold of interest. In children, in particular, the most common cause of HL is related to middle ear infections. Therefore, studies on the effect of nutrition on middle ear infections have been reported, and these studies suggest that micronutrient status and vitamin deficiency (such as vitamin A or zinc) may cause more frequent inflammation in the middle ear, leading to increased frequency of otitis media, which increases the risk of HL [69,70,71,72,73,74,75,76,77,78,79,80,81,82,83].

Most of these studies on the association between nutrition and HL have reported an increased risk of HL in certain nutritional deficiencies. However, recent studies have also shown that obesity, an excessively increased state of nutrition, increases the risk of HL [84,85,86,87,88,89,90,91,92,93]. A key hypothesis for the mechanism of the relationship between obesity and HL is that deformation of the capillary wall due to excessive fat tissue damaging the delicate inner ear system, resulting in vasoconstriction of the inner ear [94].

## 3. Studies on Nutrition and Hearing

### 3.1. Impact of Single Nutrition on Hearing (Table 1)

To date, various data have supported the possibility that the intake of certain nutrients has a particular relevance to hearing outcomes (Table 1). Some of the nutrients suggested to play a role in human hearing are vitamins, including vitamin A, B (specifically B2, B9 and B12), C and E [33,34,35,36,37,38,39,40,46,47]. Some of the minerals suggested to play a role in hearing include Mg and Se [33,36,41,42,43,44,45], although there are also studies reporting no statistically significant association between HL and Mg [95]. In another recent study, women had better hearing when retinol and vitamin B12 levels were elevated, or the intake of meat, red meat, and organ meat increased [39]. In contrast to the result in women, there was no correlation between meat intake and hearing outcomes in males, and the association between increased seafood and shellfish intake was statistically significant. The authors point out that red meat is a good source of retinol and B12. However, it is difficult to determine a definite relationship because meat provides protein or other essential nutrients. Thus, these results highlight potential interactions between nutrients, gender, possibly unknown environmental factors, genetic factors, or other individual risk factors and hearing. In some studies, iodine and iron deficiency have been reported as risk factors for HL although the exact pathophysiologic mechanism associated with them has not yet been determined [26,27]. Vitamin D deficiency can induce HL by affecting calcium metabolism and microcirculation in the cochlea or by affecting bone mineralization of the ossicles [96].

Some previous studies have shown that excessive nutrient intake is associated with hearing in humans, especially with high levels of carbohydrate, fat and cholesterol intake associated with poor hearing outcomes. It has been reported that the risk of HL may be increased in adults with a higher glycemic index (index of carbohydrate quality) and higher glycemic load (index of both quality and quantity), as well as higher total carbohydrate intake [49]. The first association between dietary fat and HL was proposed in the Mabaan tribe in Sudan where a better-than-expected cardiovascular function and hearing sensitivity were observed [50]. It was also shown that when an institutionalized patient diet was manipulated by replacing saturated fat with unsaturated fat, low hearing thresholds were observed [51]. PUFAs have been implicated in hearing results in more recent studies [36,52,53,61]. Regular fish consumption and higher intake of long-chain omega-3 PUFAs are associated with lower risk of HL in women [39]. In addition, high-density lipoprotein depletion and elevated triglycerides are associated with HL [54,55] although the relationship between HL and total cholesterol is unclear [54,56]. On the contrary, in a 2015 report in which 36,067 subjects were surveyed, an association of low fat (and low protein) diets with hearing discomfort (mean hearing thresholds), after adjustments for confounding variables such as BMI, smoking, and economic status were made, was observed [97]. In addition, it has been reported that dietary carotenoids lutein (L) and zeaxanthin (Z) play a role in maintaining optimal auditory function [98].

### 3.2. Impact of General Nutritional Status on Hearing (Table 2)

One of the most critical limitations of the single nutrition study is the inability to fully control the effect of other nutrients. This is because humans are omnivorous animals who ingest various nutrients at the same meal. Therefore, in order to overcome this, many researchers have been trying to clarify the relationship between the generalized nutrition status and various diseases by analyzing the subject’s overall eating habits, such as via the HEI [30,62]. The HEI assesses the overall quality of a person’s diet by assessing how well a person’s diet conforms to the U.S. Dietary Guidelines for Americans [99]. The use of dietary patterns allows parallel lines that better match the “real” diet, and nutrients are ingested as a combination of foods rather than individually. In other studies, dietary diversity scores were used to assess nutritional status [100]. In a Japanese community-based prospective cohort study, serum albumin and three anthropometric indices, such as body mass index (BMI), midarm circumference (MAC), and calf circumference (CC), were used to assess general nutritional status [63]. These studies suggested that lower marker values of overall nutritional status were associated with greater incidence of HL. However, the relationship between obesity and hearing status is reported in contrast to these previous studies. The current literature suggests that obesity is associated with age-related HL and sudden SNHL (SSNHL) [84,85,86,87,88,89,90,91,92]. In addition, underweight and severe obesity have been associated with an increased prevalence of HL in a cross-sectional study in a Korean population [93]. Therefore, from a review of the literature, it can be seen that various data support the possibility that dietary patterns affect hearing status and the relationship between total dietary patterns and hearing has been established in some cases (Table 2).

### 3.3. Impact of Nutrition on Pediatric Hearing (Table 3 and Table 4)

HL is the most common communication disorder that may potentiate delays in language, social, and behavioral developments in children [101]. Even children who are not severely affected may face increased risks of grade failure, behavioral problems, and interpersonal interactions, and require significant financial investment [101,102]. In addition, the specific needs for special education, amplification devices, and missed parental work have inevitable ramifications beyond the directly affected family unit, and the expected lifetime cost for a single prelingually deafened child exceeds $1,000,000 [102]. There have been many studies to identify the risk factors of pediatric HL because of its great impact both on social and economic aspects, one of which is the relationship between nutrition and HL. These studies can be divided into two major categories: (1) the study of the overall hearing status and (2) the study of HL that occurs in relation to middle ear infections.

Like adults, the relationship between general hearing status and nutrition in children can be divided into studies about single and general nutritional statuses (Table 3). One cross-sectional study assessed the HL of 0–3-month-old infants (*n* = 3386) in Nigeria. In this study, the authors defined infant malnutrition as an infant with *Z* scores less than two standard deviations below one of three indices (age-specific weight, age-specific BMI or BMI) and one of the hearing thresholds greater than 30 dB of at least one ear. Infants with poor nutritional status were significantly more likely to have severe sensory nerve impairment than infants without nutritional deficiencies [64]. Based on these results, the researchers added the weight for the length as a fourth index and could still demonstrate, using the entire study population (*n* = 6585), that early-onset hearing loss was much more prevalent among malnourished infants [65]. Single nutrition studies in pediatric populations include studies of iodine and thiamine deficiencies. One cross-sectional study showed that hearing thresholds were worse among children at risk for mild and moderate iodine deficiency and had a statistically significant correlation between hearing and urinary iodine levels at 4 kHz [66]. One randomized, placebo-controlled intervention trial (*n* = 197) with an observation period of 11 months, reported that the mildly iodine-deficient child population with higher serum thyroglobulin concentrations had significantly higher hearing thresholds in the higher frequency range (≥2 kHz) than children with lower serum thyroglobulin concentrations. Therefore, this study suggested that even if the iodine deficiency was “mild”, it was still important to promote adequate iodine intake through the salt iodination program and other methods [67]. In another study, a small case series (*n* = 11), which examined infants with thiamine deficiency resulting in pediatric intensive care unit admission, found a prevalence of HL and auditory neuropathy of between 27.3% to 45.5% among affected infants [68]. After thiamine supplementation, postintervention hearing evaluation showed improvement in 5 of 11 cases.

The most common cause of HL in the pediatric population is middle ear infections such as otitis media. Therefore, many studies concerning middle ear infection related to HL in children have been reported (Table 4). To summarize these studies, deficiencies of single micronutrients and vitamins (such as zinc, vitamin A, iron, and retinol) are associated with mild ear pathologies such as acute or chronic otitis media [69,70,71,72,73,74,75,76]. In addition, multi-micronutrient supplementation studies have reported that the frequency of HL is significantly reduced when these nutrients are supplied sufficiently [77,78,79,80,81,82,83]. In a large, well-controlled trial in Bangladeshi children, zinc supplementation showed a significant protective effect in reducing otitis media and indicates a significant importance in reducing the infection pathology of middle ear disease [76]. The precise mechanism for these is yet to be elucidated, but it is assumed that the deficiency of these nutrients affects the mucosa of the middle ear or Eustachian tube, thereby increasing the infection by increasing the inflammatory response.

### 3.4. Limitations of Previous Studies

The most frequent limitation of studies assessing the relationship between HL and nutrition is the lack of clarity about the mechanisms by which specific nutrients or general nutritional status affect HL. This is probably due to most studies to date being retrospective or cross-sectional in design. Additional studies, including animal testing, may be required to determine the mechanisms by which HL is affected by individual nutrients or nutritional status. Because the cochlea is sensitive to blood flow, the mechanisms by which nutrients affect cochlear function are likely similar to the mechanisms by which they affect various cardiovascular and neurologic diseases [58,59,60]. Systematic prospective studies and studies in animals are needed to evaluate these mechanisms.

Another limitation was that studies to date have been limited to patients with HL in a narrow sense. For example, cochlear implantation (CI) is the most successful neural prosthesis to date, with more than 220,000 implanted individuals worldwide in 2011 [103]. Education about nutritional aspects can play a significant role in the rehabilitation of patients who have undergone CI. However, no studies to date have assessed the relationship between nutrition and HL in patients who have undergone CI, suggesting the need for future research on this relationship.

### 3.5. Summary of Clinical Relevance

Studies on the relationship between HL and nutrition have reported that the incidence of HL was increased by a lack of single micro-nutrients, such as vitamins A, B, C, D and E, zinc, Mg, Se, iron and iodine. Higher intake of carbohydrates, fats, and cholesterol, and lower protein intake, corresponded to poorer hearing status, whereas higher consumption of PUFAs corresponded to better hearing status. In addition to malnutrition, obesity was reported to be a risk factor for HL. The mechanisms by which various nutritional factors influence HL have not been clearly elucidated. However, the physiological characteristics of the cochlea, which is highly influenced by blood flow and ROS, indicate that these nutritional factors may protect against HL by affecting cochlear blood flow or exhibiting antioxidative effects. Studies of the relationship between middle ear infection and nutrition in children found that lack of vitamins A, C and E, and of Zn and iron, resulted in poorer healing status due to greater vulnerability to infection.

## 4. Conclusions

We have summarized the current knowledge about the factors associated with nutrition and hearing status. The review showed that (1) various nutritional factors (such as single nutrients (vitamin A, B, C, D and E, and zinc, Mg, Se, iodine, iron, fatty acid, carbohydrate, and protein) and the generalized nutritional status) are associated with hearing status; (2) the effect of each nutritional factor may depend on other factors (such as age or gender); and (3) various nutritional factors play roles in middle ear infection (such as otitis media) in pediatric subjects. Further studies will be needed to clarify the pathophysiological mechanism of each nutrition-related factor affecting hearing. In addition, it is expected that guidelines to control nutritional factors with the aim of preventing HL can be suggested after these studies are completed.

## Figures and Tables

**Table 1 nutrients-11-00307-t001:** Studies assessing the association between hearing status and a single nutrient.

Authors and Reference	Country	Study Design	*n*	Age	Hearing Status Metric	Nutritional Factor	Outcome	Conclusion
Schieffer et al. (2017) [27]	USA	retrospective cohort study	305,339	21–90 year	identified as having hearing loss if they had at least one encounter associated with one of the following spectra of ICD-9 codes: CHL 389.0, SNHL 389.1, or hearing loss 389	iron (low serum ferritin <12.0 ng/dL)	IDA remained associated with an increased odds of combined hearing loss, adjusted OR: 2.41 (95% CI: 1.90–3.01), and also associated with increased odds of SNHL (adjusted OR: 1.82 (95% CI: 1.18–2.66)).	IDA was associated with SNHL and combined hearing loss in a population of adult patients.
Choi et al. (2014) [33]	USA	Cross-sectional	2592 (NHANES 2001–2004)	20–69 year	PTA at speech (0.5, 1, 2 and 4 kHz) and high frequencies (3, 4 and 6 kHz)	Vit. A (*β*-carotene), Vit. C, Mg	Each quartile showed dose-dependent trends with lower (better) speech-PTA and high-PTA.	(1) By using logistic regression for hearing loss, we found significant dose-dependent reductions in the odds of hearing loss across quartiles of Vit. C and E and *β*-carotene + Vit. C + Vit. E. (2) The estimated joint effects were borderline significantly larger than the sums of the individual effects (high *β*-carotene/low magnesium (−4.98%) and low *β*-carotene/high Mg (−0.80%), *p*-interaction = 0.08; high Vit. C/low Mg (−1.33%) and low Vit. C/high Mg (2.13%), *p*-interaction = 0.09). (3) Dietary intakes of antioxidants and Mg are associated with lower risks for hearing loss.
						Vit. E	Each quartile showed dose-dependent trends with lower (better) speech-PTA.	
							Each quartile except Q2 (2nd quartile, 25~50%) showed dose-dependent trends with lower (better) high-PTA.	
						*β*-carotene + Vit. C	Highest quartile had significantly lower speech-PTA (quartile 4: −14.95%; 95% CI: −20.82 to −8.65; *p*-trend <0.001) and lower high-PTA (quartile 4: −13.75%; 95% CI, −19.48 to −7.62; *p*-trend < 0.001) than did those in the lowest quartile. Lowest quartile, there was significantly lower speech-PTA in the highest quartile (quartile 4: −14.95%; 95% CI: −20.82 to −8.65; *p*-trend < 0.001).	
						*β*-carotene + Vit. C + Vit. E	Highest quartile had significantly lower speech-PTA (quartile 4: −14.81%; 95% CI, −20.80 to −8.37; *p*-trend < 0.001) and lower high-PTA (quartile 4: −13.72%; 95% CI, −20.15 to −6.77; *p*-trend < 0.001) than did those in the lowest quartile.	
Michikawa et al. (2009) [34]	Japan	community-based cross-sectional study	762	≥65 year	failure to hear a 30-dB HL signal at 1 kHz and a 40-dB HL signal at 4 kHz in the better ear in PTA	Vit. A (retinol)	Serum retinol inversely related to the prevalence of hearing impairment; adjusted OR for highest quartile compared with lowest: 0.51 (CI: 0.26–1.00; *p* = 0.03).	Increased serum levels of retinol and provitamin A carotenoids were clearly associated with a decreased prevalence of hearing impairment.
						provitamin A carotenoid (*β*-cryptoxanthin, and α- and *β*-carotenes)	Serum provitamin A family inversely related to the prevalence of hearing impairment; adjusted OR for highest quartile compared with lowest: 0.53 (CI: 0.27–1.02; *p* = 0.09).	
Gopinath et al. (2011) [35]	Blue Mountains, Sydney, Australia	Cross-sectional and 5-year longitudinal analyses	2956	≥50 year at baseline, 1997–1999, 5-year retest 2002–2004	(1) Age-related HL defined as PTA at 0.5, 1.0, 2.0 and 4.0 kHz >40 dB HL (Prevalence) (2) Age-related hearing loss defined as PTA threshold at 0.5, 1.0, 2.0 and 4.0 kHz >25 dB HL (5-year incidence) (3) Moderate or greater hearing loss defined as PTA threshold at 0.5, 1.0, 2.0 and 4.0 kHz >25 dB HL (5-year incidence) (4) Moderate or greater hearing loss defined as PTA threshold at 0.5, 1.0, 2.0 and 4.0 kHz >25 dB HL (5-year incidence)	Vit. A (retinol)	(1) Highest quintile: 47% reduced risk of hearing loss >40 dB vs. lowest quintile, adjusted OR: 0.53 (CI: 0.30–0.92; *p* = 0.04). (3) Highest quintile: no reduced risk of hearing loss >25 dB HL vs. lowest quintile, multivariable-adjusted OR 0.98 (CI: 0.71–1.37; *p* = 0.94). (4) Highest quintile: no reduced incidence of hearing loss >25 dB HL vs. lowest quintile, multivariable-adjusted OR 0.85 (CI: 0.46–1.56; *p*= 0.69).	Dietary Vit. A and Vit. E intake were significantly associated with the prevalence of hearing loss. However, dietary antioxidant intake did not increase the risk of incident hearing loss.
						Vit. A (*β*-carotene)	(1) Highest quintile compared to lowest, no reduced risk, multivariable-adjusted OR 1.17 (CI: 0.84–1.64; *p* = 0.85). (2) Highest quintile: no reduced risk vs. lowest quintile, multivariable-adjusted OR 1.13 (CI: 0.61–2.10; *p* = 0.73).	
						Vit. C	(1) Highest quintile: no reduced risk vs. lowest quintile, multivariable-adjusted OR 0.84 (CI: 0.60–1.17; *p* = 0.29). (2) Highest quintile: no reduced risk vs. lowest quintile, multivariable-adjusted OR 0.91 (CI: 0.49–1.69; *p* = 0.75).	
						Vit. E	(1) Highest quintile: no reduced risk vs. lowest quintile, multivariable-adjusted OR 1.12 (CI: 0.81–1.53; *p* = 0.30). (2) Highest quintile: no reduced risk vs. lowest quintile, multivariable-adjusted OR 0.94 (CI: 0.52–1.69; *p* = 0.70).	
						Vit. C + Vit. E	(1) A significant interaction between vitamins C and E (*p* = 0.02); subjects in the highest quintile of E but lowest quintile of C had a lower prevalence of hearing loss compared to subjects in the lowest quintile of both C and E (*p* = 0.03).	
Spankovich et al. (2011) [36]	Blue Mountains, Sydney, Australia	cross-sectional study	2111	49–99 year	(1) PTA for 250–2000 Hz (2) PTA for 3000–8000 Hz	Vit. A (retinol)	(1) Average LPTA not reliably different. (2) Average HPTA for highest quintile (mean = 44.8 dB) reliably worse than lowest quintile (mean = 41.3 dB); (*p* = 0.01).	Various nutrients with known roles in redox homeostasis and vascular health are associated with auditory function measures in a human population.
						Vit. C	(1) Highest quintile (mean = 16.8 dB) reliably different from lowest quintile (mean = 19.2 dB); (*p* = 0.005).	
						Vit. E	(1) Highest quintile (mean = 17.4 dB) different from lowest (mean = 19.9 dB); (*p* = 0.004).	
						Mg	(1) Highest quintile (mean = 17.7 dB) reliably different from lowest quintile (mean = 20.2 dB); (*p* = 0.004).	
Péneau et al. (2013) [39]	France	Cross-sectional and 13-year longitudinal analyses	1823	45–60 year at baseline	the worse ear at the following thresholds: 0.5, 1, 2 and 4 kHz	Vit. A (retinol)	Intakes of retinol (*p* = 0.058) tended to be associated with better HL in women.	Intake of retinol and Vit. B12 tended to be associated with a better HL in women.
						Vit. B12	Intakes of Vit. B12 (*p* = 0.068) tended to be associated with better HL in women.	
Shargorodsky et al. (2010) [40]	USA	Health Professionals Follow-up Study	26,273 men	40–70 year at baseline in 1986	Self-reported professionally diagnosed hearing loss, measured using the question	Vit. C, E, B12, A (*β*-carotene)	Among men 60 years and older, total folate intake was associated with a reduced risk of hearing loss; the relative risk for men ≥60 years old in the highest compared to the lowest quintile of folate intake was 0.79 (95% confidence interval 0.65–0.96).	Higher intake of Vit. C, E, B12, or beta-carotene does not reduce the risk of hearing loss in adult males. Men 60 years of age and older may benefit from higher folate intake to reduce the risk of developing hearing loss.
Joachims et al. (1993) [41]	Israel	placebo-controlled double-blind study	320	Young adult	Thresholds measured at 2,3, 4, 6 and 8 kHz before and after M16 firearm training 6 days/week × 8 weeks; ~420 shots/person, 164 dBA peak	Mg 4 g Mg granulate verum (6.7 mmol Mg aspartate) or placebo every working day during the 2-month training period	In the placebo group, the percentages of ears with PTS >25 dB at 4 kHz/6 kHz and/or 8 kHz after exposure to firearm noise were twice as high as in the Mg group.	Oral Mg-supplementation as prophylaxis against noise-induced hearing loss was effective.
Attias et al. (1994) [42]	Israel	placebo-controlled, double-blind study	300	Young adult	Thresholds measured at 2,3, 4, 6 and 8 kHz before and after M16 firearm training 6 days/week × 8 weeks; ~420 shots/person, 164 dBA peak	Mg each subject received daily an additional drink containing either 6.7 mmol (167 mg) magnesium aspartate or a similar quantity of placebo (Na-aspartate)	NIPTS was significantly more frequent and more severe in the placebo group than in the magnesium group, especially in bilateral damages.	A significant natural agent for the reduction of hearing damages in noise-exposed population.
Attias et al. (2004) [43]	Israel	double-blind manner	20 men	16–37	Thresholds measured at 1, 2, 3, 4, 6 and 8 kHz before and after a 90 dB-SL white noise × 10 min	Mg : 122 mg Mg, delivered as Mg aspartate, once/day × 10 days, or placebo	Amount of TTS reduced at 2 kHz (*p* = 0.005), 3 kHz: *p* = 0.011); prevalence of TTS reduced at 3 kHz (*p* = 0.034), 4 kHz (*p* = 0.034), 8 kHz (*p* = 0.02)	A novel, biological, natural agent for prevention and possible treatment of noise-induced cochlear damage in humans.
Weiji et al. (2004) [44]	Netherlands	randomized, double-blind, placebo-controlled study	48	Not known	PTA	Vit. C, E and Se	Patients who achieved the highest plasma concentrations of the three antioxidant micronutrients had significantly less loss of high-tone hearing.	
Chuang et al. (2007) [45]	Taiwan	case-control study	294 (control = 173, case = 121)	Not known	PTA	Se	Se was inversely associated with hearing thresholds.	Se may be a protection element on auditory function
Durga et al. (2007) [46]	Netherlands	Double-blind, randomized, placebo-controlled trial	728	Not known	3-year change in hearing thresholds, assessed as the PTA of both ears of the low (0.5-kHz, 1-kHz, and 2-kHz) and high (4-kHz, 6-kHz, and 8-kHz) frequencies.	Vit. B12 (Folic acid): Daily oral folic acid (800 mg) or placebo supplementation for 3 years	After 3 years, thresholds of the low frequencies increased by 1.0 dB (95% CI, 0.6 to 1.4 dB) in the folic acid group and by 1.7 dB (CI, 1.3 to 2.1 dB) in the placebo group (difference, −0.7 dB (CI, −1.2 to −0.1 dB); *p* = 0.020).	Folic acid supplementation slowed the decline in hearing of the speech frequencies associated with aging in a population from a country without folic acid fortification of food.
Kabagambe et al. (2018) [47]	USA	Cross-sectional study	1149 (NHANES 2003–2004)	20–69 year	PTA at 0.5, 1.0, 2.0 and 4.0 kHz was computed for each ear	Vit. B12 : Erythrocyte folate, serum Vit. B12	Compared to the 1st quartile, the ORs (95% CIs) for hearing loss were 0.87 (0.49–1.53), 0.70 (0.49–1.00), and 1.08 (0.61–1.94) for the 2nd, 3rd and 4th quartile of erythrocyte folate.	a U-shaped relationship between erythrocyte folate levels and hearing loss.
Quaranta et al. (2004) [48]	Italy	Case-control study	20	20–30 year	hearing thresholds and TTS (10 min of exposure, narrowband noise centered at 3 kHz, the bandwidth of 775 Hz, 112 dB SPL) were measured before and 8 days after treatment	Vit. B (Cyanocobalamin) Cyanocobalamin 1 mg/day × 7 days plus 5 mg on day 8), or placebo	TTS was reduced at 3 kHz (*p* < 0.001); TTS at 4 kHz was not reliably reduced (*p* = 0.061).	elevated plasma cyanocobalamin levels may reduce the risk of hearing dysfunction resulting from noise exposure in healthy, young subjects.
Gopinath et al. (2010) [49]	Blue Mountains, Sydney, Australia	population-based cross-sectional study (1997–1999 to 2002–2004)	2956	≥50 year	Hearing loss defined as the PTA of frequencies 0.5, 1.0, 2.0 and 4.0 kHz > 25 dB	Carbohydrate : dietary glycemic index (GI) and load (GL)	A higher mean dietary GI was associated with an increased prevalence of any hearing loss, comparing quintiles 1 (lowest) and 5 (highest), (multivariable-adjusted odds ratio = 1.41 (95% CI = 1.01–1.97)). Higher carbohydrate and sugar intakes were associated with incident hearing loss (*p*-trend = 0.03 and *p*-trend = 0.05, respectively).	high-GL diet was a predictor of incident hearing loss, as was a higher intake of total carbohydrate.
Dullemeijer et al. (2010) [52]	Netherlands	Cross-sectional and 3-year longitudinal analyses	720	50–70 year	PTA in the low (0.5-kHz, 1-kHz, and 2-kHz) and high (4-kHz, 6-kHz, and 8-kHz) frequencies over three years	plasma very-long-chain n-3 PUFAs	the highest quartile of plasma very-long-chain n-3 PUFA had less hearing loss in the low frequencies over three years than subjects in the lowest quartile (*p* < 0.01, ANCOVA, the difference in mean adjusted hearing thresholds= −1.2 dB).	an inverse association between plasma very-long-chain n-3 PUFAs and age-related hearing loss.
Gopinath et al. (2010) [53]	Blue Mountains, Sydney, Australia	population-based cross-sectional study (1997–1999 to 2002–2004)	2956	≥50 year	PTA of frequencies 0.5, 1.0, 2.0 and 4.0 kHz >25 decibels of hearing loss	PUFA and fish intakes	an inverse association between total n-3 PUFA intake and the prevalence of hearing loss (odds ratio (OR) per SD increase in energy-adjusted n-3 PUFAs: 0.89; 95% CI: 0.81, 0.99).	Dietary intervention with n-3 PUFAs could prevent or delay the development of age-related hearing loss
Suzuki et al. (2000) [54]	Japan	cross-sectional study	924	40–59 year	PTA of frequencies 2 KHz and 4 KHz on the better-hearing ear	serum concentrations of total cholesterol, triglyceride, and high-density lipoprotein cholesterol	As for high-density lipoprotein cholesterol, hearing levels at 2000 Hz (*p* < 0.05) and 4000 Hz (*p* < 0.01) in the high-level group were significantly better than those in the low-level group in men.	A low high-density lipoprotein cholesterol concentration is associated with hearing loss.
Evans et al. (2006) [55]	USA	cross-sectional study	40	34–73 year	PTA of frequencies at 0.25, 0.5, 1, 2, 3, 4, 6 and 8 kHz, DPOAE	triglyceride	elevated triglycerides were associated with reduced hearing.	chronic dyslipidemia associated with elevated triglycerides may reduce auditory function, short-term dietary changes may not.
Curhan et al. (2014) [61]	USA	prospective cohort study	65,215 women (Data were from Nurses’ Health Study II)	25–42 year	self-reported hearing loss	long-chain omega-3 PUFAs	In comparison with women in the lowest quintile of intake of long-chain omega-3 PUFAs, the multivariable-adjusted RR for hearing loss among women in the highest quintile was 0.85 (95% CI: 0.80, 0.91) and among women in the highest decile was 0.78 (95% CI: 0.72, 0.85) (*p*-trend < 0.001).	Regular fish consumption and higher intake of long-chain omega-3 PUFAs are associated with lower risk of hearing loss in women.
Kim et al. (2015) [97]	Korea	Cohort-based cross-sectional study	4615 (KNHNES 2009–2012)	60–80 year	PTA of frequencies at 0.5, 1, 2, 3, 4 and 6 kHz	Food intake data total energy intake, the proportion of protein, fat, carbohydrate	Low fat and protein intakes were associated with hearing discomfort (OR 0.82, 95% CI 0.71, 0.96, *p* = 0.011; OR 0.81, 95% CI 0.67, 0.96, *p* = 0.017, respectively).	low fat and protein intakes are associated with hearing discomfort in the elderly Korean population.
Wong et al. (2017) [98]	USA	Prospective study	32	18–28 year	PTA of frequencies at 0.25, 0.5, 1, 2, 3, 4, 6 and 8 kHz	Dietary carotenoids lutein (L) and zeaxanthin (Z)	L and Z status was related to many, but not all, of the pure tone thresholds we tested: 250 Hz (F(632) = 4.36, *p* < 0.01), 500 Hz (F(632) = 2.25, *p* < 0.05), 1000 Hz (F(632) = 3.22, *p* < 0.05), and 6000 Hz (F(632) = 2.56, *p* < 0.05).	The overall pattern of results is consistent with a role for L and Z in maintaining optimal auditory function.

ICD, international classification of diseases; CHL, conductive hearing loss; SNHL, sensorineural hearing loss; IDA, iron deficiency anemia; OR, odds ratio; CI, confidence interval; NHANES, National Health and Nutrition Examination Survey; PTA, pure tone average; DPOAE: distortion product otoacoustic emission; Vit., vitamin; LPTA, low-frequency of pure tone average; HPTA, high-frequency of pure tone average; dBA, decibels acoustic; SL, sensation level; SPL, sound pressure level; TTS, temporary threshold shift; Mg, magnesium; HL, hearing level; PTS, permanent threshold shifts; NIPTS, noise-induced permanent hearing threshold shifts; Se, selenium; GI, dietary glycemic index; GL, dietary glycemic load; PUFA, polyunsaturated fatty acids; KNHNES, Korean National Health and Nutrition Examination Survey; L, Dietary carotenoids lutein; Z, Dietary carotenoids zeaxanthin; ANCOVA, analysis of covariance; SD, standard deviation; RR, risk ratio.

**Table 2 nutrients-11-00307-t002:** Studies assessing the association between hearing status and general nutritional status or dietary pattern.

Authors and Reference	Country	Study Design	*n*	Age	Hearing Status Metric	Nutritional Factor	Outcome	Conclusion
Spankovich et al. (2013) [30]	USA	Cross-sectional study	2366	20–69 year (NHANES 1999–2002)	PTA of frequencies at 0.25, 0.5, 1, 2, 3, 4, 6 and 8 kHz. LFPTA (0.5, 1 and 2 kHz) and HFPTA (3, 4, 6 and 8 kHz)	HEI (HEI scores greater than 80 as “good” and scores less than 51 as “poor”.)	Controlling for age, race/ethnicity, sex, education, diabetes, and noise exposure, we found a significant negative relationship (Wald F = 6.54, df = 429; *p* ≤ 0.05) between dietary quality and thresholds at higher frequencies, where higher dietary quality was associated with lower hearing thresholds	The current findings support an association between healthier eating and lower high-frequency thresholds in adults.
Spankovich et al. (2014) [62]	USA	Cross-sectional study	2176	20–69 year (NHANES 1999–2002)	PTA of frequencies at 0.25, 0.5, 1, 2, 3, 4, 6 and 8 kHz. LFPTA (0.5, 1 and 2 kHz) and HFPTA (3, 4, 6 and 8 kHz)	HEI (HEI scores greater than 80 as “good” and scores less than 51 as “poor”.)	(1) higher (better) HEI was associated with lower (better) HFPTA (Wald F = 5.365, df = 426; *p* = 0.003). (2) lower (better) HFPTA thresholds associated with higher (better) HEI scores (Wald F = 22.438, df = 129; *p* < 0.0010). (3) The top HEI had better thresholds at individual frequencies compared to poorer HEI (at 3 kHz (Wald F = 22.453, df = 129; *p* < 0.001), 4 kHz (Wald F = 42.712, df = 129; *p* < 0.001), and 6 kHz (Wald F = 13.306, df = 129; *p* = 0.001)).	healthier diets may be associated with some small but reliable benefit with respect to HFPTA in individuals that are exposed to noise of various types and kinds.
Michikawa et al. (2016) [63]	Japan	Community-based prospective cohort study	338	≥65 year	Hearing impairment was defined as failure to hear a 30-dB hearing level signal at 1 kHz and a 40-dB signal at 4 kHz in the better ear on PTA	serum albumin, BMI, MAC, CC	Those with lower marker values had greater risk of hearing impairment than those with higher marker values (multivariable-adjusted odds ratio (aOR) = 2.18, 95% confidence interval (CI) = 1.05–4.57 for albumin ≤4.0 g/dL; aOR = 2.72, 95% CI = 1.10–6.71 for BMI <19.0 kg/m^2^). The pattern of association showed a similar tendency for MAC and CC.	Improve markers of nutritional status may help prevent age-related hearing loss in older adults.
Hwang et al. (2009) [84]	Taiwan	Prospective ross-sectional study	690	44–47 year	PTA of frequencies at 0.25, 0.5, 1, 2, 4 and 8 kHz. LFPTA (0.25, 0.5 and 1 kHz) and HFPTA (2, 4 and 8 kHz)	Obesity (WC >90 cm male and >80 cm female)	WC is independently associated with HL, but this differs by age and gender.	Central obesity was more important than BMI as a risk factor for ARHL.
Lee et al. (2015) [85]	Korea	A cross-sectional and longitudinal prospective study	1296	35–65 year	PTA—HL defined as 425 dB with no middle ear pathology	BMI, TC, TG, HDL-C, LDL-C	Elevated TC and TG levels and increased BMI are significantly associated with the prevalence of SSNHL and its prognosis, indicating that vascular compromise may play an important role in the pathogenesis of SSNHL.	Elevated TC, TG, and BMI are significantly associated with the prevalence of SSNHL.
Hwang et al. (2015) [86]	Taiwan	Retrospective cohort study	254	40–70 year	PTA: an average of 1, 2 and 4 kHz >10 dB between the affected and non-affected ear	BMI ≥25 kg/m^2^	Multivariate logistic regression analysis also showed that BMI (OR = 1.04, 95% CI = 0.964–1.131, *p* = 0.292) was not significantly associated with the recovery of SNHL for all subjects, after adjusting for all considered variables.	BMI was not significantly and independently associated with prognosis of SSNHL.
Lalwani et al. (2013) [87]	USA	Retrospective cross-sectional study	1488	12–19 year	PTA: LFPTA (0.5, 1 and 2 kHz) and HFPTA (3, 4, 6 and 8 kHz)	BMI ≥95 percentile	In multivariate analyses, obesity was associated with a 1.85-fold increase in the odds of unilateral low-frequency SNHL (95% CI: 1.10–3.13) after controlling for multiple hearing-related covariates.	Obesity in childhood is associated with higher hearing thresholds across all frequencies and an almost two-fold increase in the odds of unilateral low-frequency hearing loss.
Kang et al. (2015) [88]	Korea	Retrospective cross-sectional study	16,554	>18 year	PTA: an average of 0.5, 1, 2, 3, 4 and 6 kHz LFPTA (0.5 and 1 kHz) and MFPTA (2 and 3 kHz) and HFPTA (4 and 6 kHz)	BMI, WC, presence of metabolic syndrome	In the multivariate analysis, metabolic syndrome was associated with increased hearing thresholds in women.	Women with metabolic syndrome had higher hearing thresholds than those without.
Curhan et al. (2013) [89]	USA	Retrospective longitudinal study	68,421	25–42 year	Self-reported hearing loss	BMI, WC	Compared with women with BMI <25 kg/m^2^ the multivariate-adjusted relative risk (RR) for women with BMI ≥40 was 1.25 (95% confidence interval (CI), 1.14–1.37). Compared with women with waist circumference <71 cm, the multivariate-adjusted RR for waist circumference >88 cm was 1.27 (95% CI, 1.17–1.38).	Higher BMI and larger WC are associated with increased risk of hearing loss in women.
Kim et al. (2014) [90]	Korea	Prospective cross-sectional study	662	40–82 year	PTA	BMI, WC, visceral adipose tissue	After adjusting for age, systemic disease and other variables, a positive association between visceral adipose tissue (VAT) area and the average hearing threshold were observed in women.	Visceral adipose tissue is significantly associated with ARHL in women over 40 years.
Wu et al. (2015) [91]	Taiwan	Prospective cross-sectional study	1682	40–80 year	PTA	BMI, WC		The association between ADIPOQ and hearing threshold appears to be influenced by ADIPOR1 genotypes.
Barrenas et al. (2005) [92]	Sweden	Retrospective cohort study	245,092	0–80 year	PTA	BMI ≥25 kg/m^2^	Compared with conscripts with average body mass index, overweight was associated with 30%, obesity with 99%, and overweight if born light for gestational age with 118% higher risk of SNHL.	Increased WC was associated with a doubled risk of SNHL.
Kim et al. (2016) [93]	Korea	Cross-sectional study	61,052	≥30 year	PTA of frequencies at 0.5, 1, 2, 3, 4 and 6 kHz	BMI	Multivariate analysis showed that the odds ratios of hearing loss in the severely obese, and underweight groups, compared with the normal group, were 1.312 and 1.282, respectively.	Underweight and severe obesity were associated with an increased prevalence of hearing loss in a Korean population.
Shiraseb et al. (2016) [100]	Iran	Cross-sectional study	400	20–50 year	IVA CPT	DDS	Mean visual and auditory sustained attention showed a significant increase as the quartiles of DDS increased (*p* = 0·001).	Higher DDS is associated with better visual and auditory sustained attention.

NHANES, National Health and Nutrition Examination Survey; PTA, pure tone average; LFPTA, low-frequency PTA; MFPTA, mid-frequency PTA; HFPTA, high-frequency PTA; HL, hearing loss; SNHL, sensorineural hearing loss; OR, odds ratio; CI, confidence interval; HEI, Healthy Eating Index; BMI, body mass index; MAC, midarm circumference; CC, calf circumference.; WC, waist circumference; ADIPOR1, Adiponectin receptor 1; TC, total cholesterol; TG, triglyceride; HDL-C, high-density lipoprotein cholesterol; LDL-C, low-density lipoprotein cholesterol; ARHL, age-related hearing loss, SSNHL, sudden sensorineural hearing loss; ADIPOR, Adiponectin receptor; IVA CPT, Integrated Visual and Auditory Continuous Performance Test; DDS, Dietary diversity scores.

**Table 3 nutrients-11-00307-t003:** Studies assessing the association between pediatric hearing status and nutrition.

Authors and Reference	Country	Study Design	*n*	Age	Hearing Status Metric	Nutritional Factor	Outcome/Conclusion
Olusanya (2010) [64]	Nigeria	Cross-sectional study	3386	0–3 months	TEOAE, ABR	*Z*-score less than −2 for any of: weight-for-age, BMI-for-age, length-for-age	Infants with any undernourished physical state were significantly more likely to have severe-profound SNHL than infants without any undernourishment.
Olusanya (2011) [65]	Nigeria	Cross-sectional study	6585	0–3 months	TEOAE, ABR	*Z*-score less than −2 for any of: weight-for-age, BMI-for-age, length-for-age, weight-for-length	Undernourished infants have a significantly higher risk for early-onset permanent hearing loss.
Valeix et al. (1994) [66]	France	Cross-sectional study	1222	10 months, 2 year, 4 year	PTA of frequencies at 0.5, 1, 2, and 4 kHz.	Iodine (Urinary iodine excretion)	Hearing loss at 4000 Hz and PTA were more severe among children at risk of mild to moderate iodine deficiency; statistically significant positive correlation between hearing at 4000 Hz and urine iodine levels.
Van den Briel et al. (2001) [67]	Netherlands.	randomized, placebo-controlled intervention trial with an observation period of 11 months.	197 (iodine supplement = 97, placebo supplement = 100)	7–11 year	PTA of frequencies at 0.25, 0.5, 1, 2, 3, 4, and 6 kHz.	Iodine	In this mildly iodine-deficient child population, children with higher serum thyroglobulin concentrations had significantly higher hearing thresholds in the higher frequency range (> or = 2000 Hz) than children with lower serum thyroglobulin concentrations.
Attias et al. (2012) [68]	Israel	Case series	11	2–5 months (follow-up period: 6–8 year)	ABR, PTA	Thiamine	Human infantile thiamine deficiency may be uniquely associated with dysfunctions of the cochlea or auditory nerve, and/or auditory brainstem pathways.

TEOAE, transient evoked otoacoustic emissions; ABR, auditory brainstem response; BMI, body mass index; SNHL, sensorineural hearing loss; PTA, pure tone average.

**Table 4 nutrients-11-00307-t004:** Studies assessing the association between pediatric middle ear infection and nutrition.

Authors and Reference	Country	Study Design	*n*	Age	Nutritional Factor	Outcome/Conclusion
Brooks et al. (2005) [69]	Bangladesh	Randomized controlled trial	1621 (Zinc = 809, placebo = 52)	2–12 month	Zinc : randomly assigned zinc (70 mg) or placebo orally once weekly for 12 months.	There were significantly fewer incidents of SOM in the zinc group than the control group (relative risk 0.58, 95% CI 0.41–0.82, *p* = 0.02).
Golz et al. (2001) [70]	Israel	Observational clinical trial	880 (frequent AOM = 680, healthy = 200)	18 month–4 year	Iron	IDA children had more episodes of acute otitis media when compared with children with average levels. By increasing the hemoglobin level in these children, the frequency of the episodes of acute otitis media decreased significantly.
Tunnessen et al. (1987) [71]	USA	Longitudinal	167 (cow’s mink = 69, Iron-fortified formula=98)	6–12 month	Iron (serum ferritin levels)	No significant differences in frequency of otitis media.
D’Saouza et al. (2002) [72]	Australia	meta-analysis	1028 (Vit. A = 492, placebo = 536)	6 month–13 year	Vit. A	Vit. A does have a beneficial effect on reduction of OM (RR = 0.26 95% CI = 0.05–0.92).
Ogaro et al. (1993) [73]	Kenya	randomized controlled trial	294 (Vit. A = 146, placebo = 148)	<5 year	Vit. A	Lower rates of OM in Vit. A supplementation (RR 0.22, 95% CI = 0.06–0.90, *p* = 0.03).
Lasisi (2008) [74]	Nigeria	Case-control study	316 (case = 264, control = 52)	6 month–11 year	Vit. A(retinol)	Retinol supplementation is a possible nutritional approach to control SOM (*p* = 0.000).
Sarmila et al. (2001) [75]	India	Case-control study	300 (case = 150, control = 150)	5–14 year	Vit. A	Increased frequency of Vit. A deficiency with CSOM.
Durand et al. (1997) [76]	USA	prospective, observational study	200	3–5 year	Vit. A	The status of Vit. A and related compounds in children appeared to have no effect on the incidence of otitis media.
Linday et al. (2002) [77]	USA	Pilot study	8	0.8–4.4 year	Vit. A, Se	Fewer days of antibiotics for OM during Vit. A and Se supplementation.
Cemek et al. (2005) [78]	Turkey	Comparative study	50 (AOM = 23, AT = 27)	2–7 year	Vit. A (*β*-carotene, retinol), C, E	AOM and AT tissue may react differently to oxidative stress.
Omonov (1997) [79]	Uzbekistan	Cross-sectional study	48	3–15 year	Multi-vitamin, minerals	Reduced levels of iron, zinc, selenium and bromine in children with CSOM.
Jones et al. (2006) [80]	Australia	Longitudinal historical control study	15	4–11 year	Multi-vitamin, minerals	Mean antibiotic prescriptions for OM decreased from seven to 1 per month.
Daly et al. (1999) [81]	USA	Longitudinal study	596	0–6 month	Multi-vitamin, minerals	Among prenatal exposures, only high prenatal dietary Vit. C intake was significantly inversely related to early AOM with univariate but not multivariate analysis.
Dobó et al. (1998) [82]	Hungary	randomized double-blind trial	625 (multi-vitamin = 323, trace elements = 302)	2–6 year	Folic acid-containing multi-vitamin	Higher incidence of AOM in the multi-vitamin group.
Karabaev (1997) [83]	Russia	Longitudinal study	Not known	6 month–15 year	Vit. A (retinol), C (ascorbic acid), E (α-tocopherol)	The beneficial effect in SOM.

OM, otitis media; SOM, suppurative otitis media; AOM, acute otitis media; IDA, iron-deficiency anemia; Vit., vitamin; CSOM, chronic suppurative otitis media; Se, selenium; AT, acute tonsillitis; RR, risk ratio; CI, confidence interval.

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
