# Peer review of "Association of Nutritional Factors with Hearing Loss"

_nutrients, 2019, doi:10.3390/nu11020307_

Round 1
Reviewer 1 Report
The paper is interesting, but sometimes it is repetitive and this makes it not very fluent. Many parts must be rewritten.
The first sentence of the introduction doesn't seem correct and the reference 1 seems to be related only to elder patients.
The paper in many parts is pleonastic and repetitive(for example, the issue of iodic depletion is too stressed, both in adults and in the pediatric part). In our opinion it could be only mentioned in the introduction, sincethere are specific related tables. In the same way, in paragraph 3.2 (Impact of General Nutritional Status on Hearing) the major point can be summarized by saying that a good nutritional status, with anthropometric indices in the normal range (neither abov as in obesity, nor below as malnutrition) is associated with lower incidence of hearing impairment, at all ages; even in this case, we found detailed tables at the end of the paragraph.
Author Response
Q1. The paper is interesting, but sometimes it is repetitive and this makes it not very fluent. Many parts must be rewritten.
A) We have revised our manuscript to remove redundancies. Changes in the manuscript have been highlighted.
Q2. The first sentence of the introduction doesn't seem correct and the reference 1 seems to be related only to elder patients.
A) We agree that the reference in the first sentence of the Introduction describes hearing loss in elderly individuals, whereas the sentence itself describes hearing loss in general. We have therefore changed the reference from:
1. Bagai, A.; Thavendiranathan, P.; Detsky, A.S. Does this patient have hearing impairment? JAMA. 2006, 295, 416-428.
to
1. Nash, S.D.; Cruickshanks, K.J.; Klein, R.; Klein, B.E.; Nieto, F.J.; Huang, G.H.; Pankow, J.S.; Tweed, T.S. The prevalence of hearing impairment and associated risk factors: the Beaver Dam Offspring Study. Arch Otolaryngol Head Neck Surg. 2011, 137, 432-439.
Q3. The paper in many parts is pleonastic and repetitive(for example, the issue of iodic depletion is too stressed, both in adults and in the pediatric part). In our opinion it could be only mentioned in the introduction, sincethere are specific related tables. In the same way, in paragraph 3.2 (Impact of General Nutritional Status on Hearing) the major point can be summarized by saying that a good nutritional status, with anthropometric indices in the normal range (neither abov as in obesity, nor below as malnutrition) is associated with lower incidence of hearing impairment, at all ages; even in this case, we found detailed tables at the end of the paragraph.
A) We agree with your suggestions. The most important consideration for us when preparing this review was whether nutrition or nutritional status could be a factor affecting hearing. Surprisingly, various nutrients and nutritional factors have been shown to cause either hearing loss or improvement. Our aim was to provide researchers interested in hearing loss and nutrition with an overview of these numerous research findings, and to integrate a lot of related information. However, no definite data have been published on the causative mechanism of each nutrient or nutritional factor. Most of the literature assumes that the cochlea, the organ controlling hearing, is sensitive to blood flow and that each nutrient or nutritional state affects hearing by affecting cochlear blood flow. In addition, some nutrients showing antioxidant effects have been shown to inhibit the production of ROS, thereby minimizing hearing loss by preventing hair cell damage. [Please check that the rewriting of the previous sentence retains the intended meaning] These mechanisms have been reported to be important for all people, without distinction between children and adults. However, most data on hearing loss associated with otitis media derive from pediatric studies, so the above discussion is reviewed in another section (for example: in section 3.3.).
In most of the literature we reviewed, each researcher assumed that nutrients and nutritional status would act as a mechanism similar to that mentioned above for the mechanisms that have a specific effect on hearing. Therefore, our review article repeatedly refers to these areas. However, based on your suggestions, we have actively tried to reduce this repetition. These amendments can be easily seen in the track-changes version of the manuscript that we have provided.
Reviewer 2 Report
This article considers the role of nutritional factors on hearing loss in adults and children, it summarizes more than 100 articles on the subject. The material is clearly presented in several tables, which are accompanied by the general description of the findings in literature, their discussion and conclusions.
The authors mention that a lot of described factors are non-specific, e.g. also related to cardiovascular and cerebrovascular diseases. This is not surprising because the cochlea is dependent on good vascularization, besides good auditory nerve function is necessary. In general, except for the role of zinc, the findings in literature are in line with the necessity of good vascular and nervous system functioning. Thus, they are non-specific for the auditory system. I think, more attention should be given to this aspect. On the contrary, it remains unclear what specific nutritional factors are important for good hearing. This can be stated as a perspective.
Hearing loss in the article is considered in a narrow sense, not including patients after cochlear implantation. However, undoubtedly nutritional factors are important for the rehabilitation of the implanted patients. This aspect could also be developed, at least in some phrases or indicated as a perspective.
Author Response
Q) The authors mention that a lot of described factors are non-specific, e.g. also related to cardiovascular and cerebrovascular diseases. This is not surprising because the cochlea is dependent on good vascularization, besides good auditory nerve function is necessary. In general, except for the role of zinc, the findings in literature are in line with the necessity of good vascular and nervous system functioning. Thus, they are non-specific for the auditory system. I think, more attention should be given to this aspect. On the contrary, it remains unclear what specific nutritional factors are important for good hearing. This can be stated as a perspective.
A) We fully agree with this reviewer. Few studies to date, including animal experiments, have identified the exact factors that affect specific nutrients or nutritional status. Rather, most studies have only speculated that certain nutrients may have a positive effect on blood vessels supplying the cochlea, the organ most closely related to hearing, thereby reducing the risk of hearing loss. The cochlea was found to respond sensitively to changes in blood flow, suggesting that nutrients would have similar effects on cardiovascular and other cerebral nerves. We were therefore unable to precisely determine the mechanism of action of each nutrient. We believe that this also is a limitation of our manuscript. We added additional sentences to the Discussion section, as shown in the answer to Q2, below.
Many of the papers we reviewed reported that certain antioxidants (such as vitamins A and C) and several trace elements (such as Zn, Se, Mg, and Fe) protect against hearing loss. Overall nutritional status, such as Healthy Eating Index (HEI), was also reported to be related to hearing loss. These findings suggest that the risk of hearing loss may be reduced by integrating these factors. Other factors that may protect against hearing loss include better overall nutritional status and supplementation with antioxidant vitamins and certain micronutrients. These results may be useful in nutritional consultation for high-risk individuals with hearing loss, including workers at risk of exposure to noise and elderly individuals at risk of nutritional imbalance.
Q) Hearing loss in the article is considered in a narrow sense, not including patients after cochlear implantation. However, undoubtedly nutritional factors are important for the rehabilitation of the implanted patients. This aspect could also be developed, at least in some phrases or indicated as a perspective.
A) Cochlear implantation (CI) is an important method of treating patients with hearing loss. We also believe that nutritional factors play a significant role in the rehabilitation of patients who have undergone CI. To our knowledge, however, no studies have reported that nutritional factors are significant for improvement of hearing loss following CI. Rather, most studies related to CI have focused on the therapeutic results of CI itself and on factors predicting outcomes. Despite our inability to identify studies of the relationship between nutritional status and CI, we believe that future studies should address this relationship. We have therefore included the following sentences.
3.4. Limitations of Previous Studies
The most frequent limitation of studies assessing the relationship between HL and nutrition is the lack of clarity about the mechanisms by which specific nutrients or general nutritional status affect HL. This is probably due to most studies to date being retrospective or cross-sectional in design. Additional studies, including animal testing, may be required to determine the mechanisms by which HL is affected by individual nutrients or nutritional status. Because the cochlea is sensitive to blood flow, the mechanisms by which nutrients affect cochlear function are likely similar to the mechanisms by which they affect various cardiovascular and neurologic diseases [58-60]. Systematic prospective studies and studies in animals are needed to evaluate these mechanisms.
Another limitation was that studies to date have been limited to patients with HL in a narrow sense. For example, cochlear implantation (CI) is the most successful neural prosthesis to date, with more than 220,000 implanted individuals worldwide in 2011 [103]. Education about nutritional aspects can play a significant role in the rehabilitation of patients who have undergone CI. However, no studies to date have assessed the relationship between nutrition and HL in patients who have undergone CI, suggesting the need for future research on this relationship.
103. Cosetti, M.K.; Waltzman, S.B. Cochlear implants: current status and future potential. Expert Rev Med Devices. 2011, 8, 389-401.
Reviewer 3 Report
In this review of the literature the authors focus on the effects of nutrition on hearing. The paper is well written and easy to read. I have some recomendations:
1) You can omit the abbreviation "NHANES" for the US National Health and Nutrition Examination Survey since this is not mentioned again throughout the manuscript
2) pag 2, line 72, I suggest removing "our" from "our increasingly noisy environment"
3) there are some english grammar mistakes in the manuscript. One example is on pag 2 line 76 "nutrition play" should be "nutrition plays"
4) In many studies available in the literature that focused on the role of antioxidants on hearing loss (see page 2 lines 86-93), the antioxidants were administered at very high doses for a short time before, during and after the acoustic trauma. This maximizes the protective effects of the drug since high blood concentration of antioxidant drug is present when noise exposure occurs. This should be discussed in more details.
5) page 3, line 104: HEI (Healthy Eating Index). Abbreviation should follow the full text, not the opposite
6) I'd recommend adding a discussion paragraph in which the authors discuss the results of their review of the literature more in details, also trying to summarize their findings in a more comprehensive way.
Author Response
Q1) You can omit the abbreviation "NHANES" for the US National Health and Nutrition Examination Survey since this is not mentioned again throughout the manuscript
A) This abbreviation has been omitted.
Q2) pag 2, line 72, I suggest removing "our" from "our increasingly noisy environment"
A) The word “our” has been deleted.
Q3) there are some english grammar mistakes in the manuscript. One example is on pag 2 line 76 "nutrition play" should be "nutrition plays"
A) The entire manuscript has again been proofread by an English editing service.
Q4) In many studies available in the literature that focused on the role of antioxidants on hearing loss (see page 2 lines 86-93), the antioxidants were administered at very high doses for a short time before, during and after the acoustic trauma. This maximizes the protective effects of the drug since high blood concentration of antioxidant drug is present when noise exposure occurs. This should be discussed in more details.
A) We have made the following changes to the manuscript:
Therefore, several studies have been reported on the relationship between HL and vitamin A [33-39], C [33,35-37,40], and E in humans [33-37,40]. Magnesium (Mg) has also been reported to reduce HL through synergistic effects with vitamins [33,36,41-43]. According to these studies, free radical scavengers, such as vitamins A, C, and E, act in synergy with Mg to reduce changes in hearing thresholds more reliably than treatment with any single agent. Therefore, it was suggested that higher intakes of antioxidants or magnesium, or antioxidant and magnesium in combination, appear to be associated with a lower risk of HL. Selenium (Se) has also been suggested to play a role in hearing [44,45]. In addition, there is a report that the lack of vitamin B can increase the risk of HL [39,46-48] while other experimental studies in animal models support the conclusion that antioxidants improve potential free radical damage associated with micronutrient deficiency.
Magnesium (Mg) has also been reported to reduce HL through synergistic effects with vitamins [33,36,41-43]. These findings suggest that free radical scavengers, such as vitamins A, C, and E, act in synergy with Mg to reduce changes in hearing thresholds more reliably than treatment with any single agent. Therefore, higher intake of antioxidants and/or magnesium may be associated with a lower risk of HL. Selenium (Se) may also play a role in hearing [44,45]. Moreover, a lack of vitamin B has been reported to increase the risk of HL [39,46-48], whereas studies in animal models indicate that antioxidants reduce potential free radical damage associated with micronutrient deficiency. Exposure to intense noise over time activates morphological, physical and mechanical mechanisms that can cause cochlear damage and result in HL [31]. Metabolic and molecular mechanisms may also be responsible for this type of lesion [32]. Although the exact mechanisms required for the induction of temporary HL are unclear, recent studies indicate that the formation of reactive oxygen species (ROS) and oxidative stress are the main metabolic causes of temporary HL [43,48]. Permanent HL of cochlear origin is thought to be caused by the death of hair cells, both internal and external, primary afferent neurons, or both [41,42]. ROS are molecules characterized by the presence of unpaired electrons; they are always present in the body and participate in homeostasis and important signaling pathways. However, imbalances in the endogenous antioxidant system can increase ROS concentrations to toxic levels, causing cell death due to damage to cell membranes, cytoplasm and mitochondria. In particular, excess free radicals in the cells of the cochlear sensory epithelium, spiral ganglion neurons and cochlear blood vessels can play an important role in the onset of HL. Excess ROS, induced by intense noise exposure and toxic drugs, is a key factor in the pathogenesis of HL, along with other stresses and aging [33,35-37,44,45]. Several studies have shown that loud noise induces unnecessary ROS production and the overproduction of certain cells in the cochlea resulting in cell damage and noise-induced HL. Under these conditions, high blood concentrations of antioxidants could inhibit the production of ROS, suggesting that antioxidants reduce cell damage by suppressing ROS production in hair cells exposed to noise [31-33].
Q5) page 3, line 104: HEI (Healthy Eating Index). Abbreviation should follow the full text, not the opposite
A) This sentence has been revised:
In another study, HEI (Healthy Eating Index) was used as a measure of total dietary value to utilize dietary patterns as an alternative method to investigate the relationship between nutrition and hearing considering the limitations of single nutritional analysis [30,62].
Another study employed the Healthy Eating Index (HEI) to measure total dietary value, enabling dietary patterns to be used as an alternative method to investigate the relationship between nutrition and hearing, overcoming the limitations of analyses of single nutrients [30,62].
Q6) I'd recommend adding a discussion paragraph in which the authors discuss the results of their review of the literature more in details, also trying to summarize their findings in a more comprehensive way.
A) We added a paragraph entitled "Summary of Clinical Relevance" to the end of the Discussion section. This paragraph includes a detailed summary of the results of our literature review:
3.5. Summary of Clinical Relevance
Studies on the relationship between HL and nutrition have reported that the incidence of HL was increased by a lack of single micro-nutrients, such as vitamins A, B, C, D, and E, zinc, Mg, Se, iron, and iodine. Higher intake of carbohydrates, fats, and cholesterol, and lower protein intake, corresponded to poorer hearing status, whereas higher consumption of PUFAs corresponded to better hearing status. In addition to malnutrition, obesity was reported to be a risk factor for HL. The mechanisms by which various nutritional factors influence HL have not been clearly elucidated. However, the physiological characteristics of the cochlea, which is highly influenced by blood flow and ROS, indicate that these nutritional factors may protect against HL by affecting cochlear blood flow or exhibiting antioxidative effects. Studies of the relationship between middle ear infection and nutrition in children found that lack of vitamins A, C, and E, and of Zn and iron, resulted in poorer healing status due to greater vulnerability to infection.
Round 2
Reviewer 1 Report
In "Evidence that nutrition is a factor affecting HL" we found very interesting the discussion about ROS, not present in the previous version of the paper. Paragraph 3.2 has been made lighter and more fluid.The paragraph 3.4, "Limitations of previous studies" is appropriate and interesting. Paragraph, "Summary of clinical relevance" is now well written.
Minor revision
In lines 32-33 the phrase on the frequency of ear disorders is still present. We are still of the opinion that it should be eliminated.
In lines 304-308 we found exactly the same sentence written twice.
Author Response
Q1. In lines 32-33 the phrase on the frequency of ear disorders is still present. We are still of the opinion that it should be eliminated.
A) We agree with your suggestions. So, we have revised our manuscript as follow;
1. Introduction
Hearing loss (HL), caused by partial or complete dysfunction of the auditory pathway from the external ear to the cerebral auditory cortex, is the most important and frequent symptom of ear disorders [1]. According to the 2012 US National Health and Nutrition Examination Survey (NHANES), approximately 15% of US adults reported hearing abnormalities, with a higher prevalence among men and older individuals [2,3]. HL is a major public health problem and has been recently ranked as the fifth leading cause of years lived with disability, even higher than many other chronic diseases such as diabetes, dementia, and chronic obstructive pulmonary disease. Moreover, according to the World Health Organization, the incidence of auditory disorders is increasing at an alarming rate without much concern and perception by society and public health officials [4,5]. Because HL affects the quality of life and affects the communication and relationships with others, it is related to the social cost as well as economic costs of medical treatment [6].
1. Introduction
Hearing loss (HL), caused by partial or complete dysfunction of the auditory pathway from the external ear to the cerebral auditory cortex, is the most important and frequent symptom of ear disorders [1]. HL is a major public health problem and has been recently ranked as the fifth leading cause of years lived with disability, even higher than many other chronic diseases such as diabetes, dementia, and chronic obstructive pulmonary disease. Moreover, according to the World Health Organization, the incidence of auditory disorders is increasing at an alarming rate without much concern and perception by society and public health officials [2-5]. Because HL affects the quality of life and affects the communication and relationships with others, it is related to the social cost as well as economic costs of medical treatment [6].
Q2. In lines 304-308 we found exactly the same sentence written twice.
A) This is our typo. We have revised these parts.
3.5. Summary of Clinical Relevance
Studies on the relationship between HL and nutrition have reported that the incidence of HL was increased by a lack of single micro-nutrients, such as vitamins A, B, C, D, and E, zinc, Mg, Se, iron, and iodine. Studies on the relationship between HL and nutrition have reported that the incidence of HL was increased by a lack of single micro-nutrients, such as vitamins A, B, C, D, and E, zinc, Mg, Se, iron, and iodine. Higher intake of carbohydrates, fats, and cholesterol, and lower protein intake, corresponded to poorer hearing status, whereas higher consumption of PUFAs corresponded to better hearing status. In addition to malnutrition, obesity was reported to be a risk factor for HL. The mechanisms by which various nutritional factors influence HL have not been clearly elucidated. However, the physiological characteristics of the cochlea, which is highly influenced by blood flow and ROS, indicate that these nutritional factors may protect against HL by affecting cochlear blood flow or exhibiting antioxidative effects. Studies of the relationship between middle ear infection and nutrition in children found that lack of vitamins A, C, and E, and of Zn and iron, resulted in poorer healing status due to greater vulnerability to infection.
Studies on the relationship between HL and nutrition have reported that the incidence of HL was increased by a lack of single micro-nutrients, such as vitamins A, B, C, D, and E, zinc, Mg, Se, iron, and iodine. Higher intake of carbohydrates, fats, and cholesterol, and lower protein intake, corresponded to poorer hearing status, whereas higher consumption of PUFAs corresponded to better hearing status. In addition to malnutrition, obesity was reported to be a risk factor for HL. The mechanisms by which various nutritional factors influence HL have not been clearly elucidated. However, the physiological characteristics of the cochlea, which is highly influenced by blood flow and ROS, indicate that these nutritional factors may protect against HL by affecting cochlear blood flow or exhibiting antioxidative effects. Studies of the relationship between middle ear infection and nutrition in children found that lack of vitamins A, C, and E, and of Zn and iron, resulted in poorer healing status due to greater vulnerability to infection.